

# Predicting judicial decisions of the European Court of Human Rights: a Natural Language Processing perspective

Nikolaos Aletras[1,2], Dimitrios Tsarapatsanis[3], Daniel Preoţiuc-Pietro[4,5] and Vasileios Lampos[2]

[1] Amazon.com, Cambridge, United Kingdom
[2] Department of Computer Science, University College London, University of London, London, United Kingdom
[3] School of Law, University of Sheffield, Sheffield, United Kingdom
[4] Positive Psychology Center, University of Pennsylvania, Philadelphia, United States
[5] Computer & Information Science, University of Pennsylvania, Philadelphia, United States

## ABSTRACT

Recent advances in Natural Language Processing and Machine Learning provide us with the tools to build predictive models that can be used to unveil patterns driving judicial decisions. This can be useful, for both lawyers and judges, as an assisting tool to rapidly identify cases and extract patterns which lead to certain decisions. This paper presents the first systematic study on predicting the outcome of cases tried by the European Court of Human Rights based solely on textual content. We formulate a binary classification task where the input of our classifiers is the textual content extracted from a case and the target output is the actual judgment as to whether there has been a violation of an article of the convention of human rights. Textual information is represented using contiguous word sequences, i.e., N-grams, and topics. Our models can predict the court's decisions with a strong accuracy (79% on average). Our empirical analysis indicates that the formal facts of a case are the most important predictive factor. This is consistent with the theory of legal realism suggesting that judicial decision-making is significantly affected by the stimulus of the facts. We also observe that the topical content of a case is another important feature in this classification task and explore this relationship further by conducting a qualitative analysis.

# INTRODUCTION

In his prescient work on investigating the potential use of information technology in the legal domain, Lawlor surmised that computers would one day become able to analyse and predict the outcomes of judicial decisions (*Lawlor, 1963*). According to Lawlor, reliable prediction of the activity of judges would depend on a *scientific understanding* of the ways that the law and the facts impact on the relevant decision-makers, i.e., the judges. More than fifty years later, the advances in Natural Language Processing (NLP) and Machine Learning (ML) provide us with the tools to automatically analyse legal materials, so as to build successful predictive models of judicial outcomes.

Corresponding author
Nikolaos Aletras,
nikos.aletras@gmail.com

In this paper, our particular focus is on the automatic analysis of cases of the European Court of Human Rights (*ECtHR* or *Court*). The ECtHR is an international court that rules on individual or, much more rarely, State applications alleging violations by some State Party of the civil and political rights set out in the European Convention on Human Rights (*ECHR* or *Convention*). Our task is to predict whether a particular Article of the Convention has been violated, given textual evidence extracted from a case, which comprises of specific parts pertaining to the facts, the relevant applicable law and the arguments presented by the parties involved. Our main hypotheses are that (1) the textual content, and (2) the different parts of a case are important factors that influence the outcome reached by the Court. These hypotheses are corroborated by the results. Our work lends some initial plausibility to a text-based approach with regard to ex ante prediction of ECtHR outcomes on the assumption that the text extracted from published judgments of the Court bears a sufficient number of similarities with, and can therefore stand as a (crude) proxy for, applications lodged with the Court as well as for briefs submitted by parties in pending cases. We submit, though, that full acceptance of that reasonable assumption necessitates more empirical corroboration. Be that as it may, our more general aim is to work under this assumption, thus placing our work within the larger context of ongoing empirical research in the theory of adjudication about the determinants of judicial decision-making. Accordingly, in the discussion we highlight ways in which automatically predicting the outcomes of ECtHR cases could potentially provide insights on whether judges follow a so-called *legal model* (*Grey, 1983*) of decision making or their behavior conforms to the *legal realists' theorization* (*Leiter, 2007*), according to which judges primarily decide cases by responding to the stimulus of the facts of the case.

We define the problem of the ECtHR case prediction as a binary classification task. We utilise textual features, i.e., N-grams and topics, to train Support Vector Machine (SVM) classifiers (*Vapnik, 1998*). We apply a linear kernel function that facilitates the interpretation of models in a straightforward manner. Our models can reliably predict ECtHR decisions with high accuracy, i.e., 79% on average. Results indicate that the 'facts' section of a case best predicts the actual court's decision, which is more consistent with legal realists' insights about judicial decision-making. We also observe that the topical content of a case is an important indicator whether there is a violation of a given Article of the Convention or not.

Previous work on predicting judicial decisions, representing disciplinary backgrounds in political science and economics, has largely focused on the analysis and prediction of judges' votes given non textual information, such as the nature and the gravity of the crime or the preferred policy position of each judge (*Kort, 1957*; *Nagel, 1963*; *Keown, 1980*; *Segal, 1984*; *Popple, 1996*; *Lauderdale & Clark, 2012*). More recent research shows that information from texts authored by *amici curiae*[1] improves models for predicting the votes of the US Supreme Court judges (*Sim, Routledge & Smith, 2015*). Also, a text mining approach utilises sources of metadata about judge's votes to estimate the degree to which those votes are about common issues (*Lauderdale & Clark, 2014*). Accordingly, this paper presents the first systematic study on predicting the decision outcome of cases tried at a major international court by mining the available textual information.

[1] An *amicus curiae* (friend of the court) is a person or organisation that offers testimony before the Court in the context of a particular case without being a formal party to the proceedings.

Overall, we believe that building a text-based predictive system of judicial decisions can offer lawyers and judges a useful assisting tool. The system may be used to rapidly identify cases and extract patterns that correlate with certain outcomes. It can also be used to develop prior indicators for diagnosing potential violations of specific Articles in lodged applications and eventually prioritise the decision process on cases where violation seems very likely. This may improve the significant delay imposed by the Court and encourage more applications by individuals who may have been discouraged by the expected time delays.

## MATERIALS AND METHODS

### European Court of Human Rights

The ECtHR is an international court set up in 1959 by the ECHR. The court has jurisdiction to rule on the applications of individuals or sovereign states alleging violations of the civil and political rights set out in the Convention. The ECHR is an international treaty for the protection of civil and political liberties in European democracies committed to the rule of law. The treaty was initially drafted in 1950 by the ten states which had created the Council of Europe in the previous year. Membership in the Council entails becoming party to the Convention and all new members are expected to ratify the ECHR at the earliest opportunity. The Convention itself entered into force in 1953. Since 1949, the Council of Europe and thus the Convention have expanded significantly to embrace forty-seven states in total, with a combined population of nearly 800 million. Since 1998, the Court has sat as a full-time court and individuals can apply to it directly, if they can argue that they have voiced their human rights grievance by exhausting all effective remedies available to them in their domestic legal systems before national courts.

### *Case processing by the court*

The vast majority of applications lodged with the Court are made by individuals. Applications are first assessed at a prejudicial stage on the basis of a list of admissibility criteria. The criteria pertain to a number of procedural rules, chief amongst which is the one on the exhaustion of effective domestic remedies. If the case passes this first stage, it can either be allocated to a single judge, who may declare the application inadmissible and strike it out of the Court's list of cases, or be allocated to a Committee or a Chamber. A large number of the applications, according to the court's statistics fail this first admissibility stage. Thus, to take a representative example, according to the Court's provisional annual report for the year 2015,[2] 900 applications were declared inadmissible or struck out of the list by Chambers, approximately 4,100 by Committees and some 78,700 by single judges. To these correspond, for the same year, 891 judgments on the merits. Moreover, cases held inadmissible or struck out are not reported, which entails that a text-based predictive analysis of them is impossible. It is important to keep this point in mind, since our analysis was solely performed on cases retrievable through the electronic database of the court, HUDOC.[3] The cases analysed are thus the ones that have already passed the first admissibility stage,[4] with the consequence that the Court decided on these cases' merits under one of its formations.

[2]ECHtR provisional annual report for the year 2015: http://www.echr.coe.int/Documents/Annual_report_2015_ENG.pdf.

[3]HUDOC ECHR Database: http://hudoc.echr.coe.int/.

[4]Nonetheless, not all cases that pass this first admissibility stage are decided in the same way. While the individual judge's decision on admissibility is final and does not comprise the obligation to provide reasons, a Committee deciding a case may, by unanimous vote, declare the application admissible and render a judgment on its merits, if the legal issue raised by the application is covered by well-established case-law by the Court.

### Main premise

Our main premise is that published judgments can be used to test the possibility of a text-based analysis for ex ante predictions of outcomes on the assumption that there is enough similarity between (at least) certain chunks of the text of published judgments and applications lodged with the Court and/or briefs submitted by parties with respect to pending cases. Predictive tasks were based on the text of published judgments rather than lodged applications or briefs simply because we did not have access to the relevant data set. We thus used published judgments as proxies for the material to which we do not have access. This point should be borne in mind when approaching our results. At the very least, our work can be read in the following hypothetical way: if there is enough similarity between the chunks of text of published judgments that we analyzed and that of lodged applications and briefs, then our approach can be fruitfully used to predict outcomes with these other kinds of texts.

### Case structure

The judgments of the Court have a distinctive structure, which makes them particularly suitable for a text-based analysis. According to Rule 74 of the Rules of the Court,[5] a judgment contains (among other things) an account of the procedure followed on the national level, the facts of the case, a summary of the submissions of the parties, which comprise their main legal arguments, the reasons in point of law articulated by the Court and the operative provisions. Judgments are clearly divided into different sections covering these contents, which allows straightforward standardisation of the text and consequently renders possible text-based analysis. More specifically, the sections analysed in this paper are the following:

[5] Rules of ECtHR, http://www.echr.coe.int/Documents/Rules_Court_ENG.pdf.

- **Procedure:** This section contains the procedure followed before the Court, from the lodging of the individual application until the judgment was handed down.
- **The facts:** This section comprises all material which is not considered as belonging to points of law, i.e., legal arguments. It is important to stress that the facts in the above sense do not just refer to actions and events that happened in the past as these have been formulated by the Court, giving rise to an alleged violation of a Convention article. The 'Facts' section is divided in the following subsections:

  - **The circumstances of the case:** This subsection has to do with the factual background of the case and the procedure (typically) followed before domestic courts before the application was lodged by the Court. This is the part that contains materials relevant to the individual applicant's story in its dealings with the respondent state's authorities. It comprises a recounting of all actions and events that have allegedly given rise to a violation of the ECHR. With respect to this subsection, a number of crucial clarifications and caveats should be stressed. To begin with, the text of the 'Circumstances' subsection has been formulated by the Court itself. As a result, it should not always be understood as a neutral mirroring of the factual background of the case. The choices made by the Court when it comes to formulations of the facts incorporate implicit or explicit judgments to the effect that some facts are more

relevant than others. This leaves open the possibility that the formulations used by the Court may be tailor-made to fit a specific preferred outcome. We openly acknowledge this possibility, but we believe that there are several ways in which it is mitigated. First, the ECtHR has limited fact-finding powers and, in the vast majority of cases, it defers, when summarizing the factual background of a case, to the judgments of domestic courts that have already heard and dismissed the applicants' ECHR-related complaint (*Leach, Paraskeva & Uelac, 2010*; *Leach, 2013*). While domestic courts do not necessarily hear complaints on the same legal issues as the ECtHR does, by virtue of the incorporation of the Convention by all States Parties (*Helfer, 2008*), they typically have powers to issue judgments on ECHR-related issues. Domestic judgments may also reflect assumptions about the relevance of various events, but they also provide formulations of the facts that have been validated by more than one decision-maker. Second, the Court cannot openly acknowledge any kind of bias on its part. This means that, on their face, summaries of facts found in the 'Circumstances' section have to be at least framed in as neutral and impartial a way as possible. As a result, for example, clear displays of impartiality, such as failing to mention certain crucial events, seem rather improbable. Third, a cursory examination of many ECtHR cases indicates that, in the vast majority of cases, parties do not seem to dispute the facts themselves, as contained in the 'Circumstances' subsection, but only their legal significance (i.e., whether a violation took place or not, given those facts). As a result, the 'Circumstances' subsection contains formulations on which, in the vast majority of cases, disputing parties agree. Last, we hasten to add that the above three kinds of considerations do not logically entail that other forms of non-outright or indirect bias in the formulation of facts are impossible. However, they suggest that, in the absence of access to other kinds of textual data, such as lodged applications and briefs, the 'Circumstances' subsection can reasonably perform the function of a (sometimes crude) proxy for a textual representation of the factual background of a case.

  – **Relevant law:** This subsection of the judgment contains all legal provisions other than the articles of the Convention that can be relevant to deciding the case. These are mostly provisions of domestic law, but the Court also frequently invokes other pertinent international or European treaties and materials.

- **The law:** The law section considers the merits of the case, through the use of legal argument. Depending on the number of issues raised by each application, the section is further divided into subsections that examine individually each alleged violation of some Convention article (see below). However, the Court in most cases refrains from examining all such alleged violations in detail. Insofar as the same claims can be made by invoking more than one article of the Convention, the Court frequently decides only those that are central to the arguments made. Moreover, the Court frequently refrains from deciding on an alleged violation of an article, if it overlaps sufficiently with some other violation it has already decided on.

  – **Alleged violation of article *x*:** Each subsection of the judgment examining alleged violations in depth is divided into two sub-sections. The first one contains the *Parties'*

## PROCEDURE

1. The case originated in an application (no. 35355/08) against the Republic of Bulgaria lodged with the Court under Article 34 of the Convention for the Protection of Human Rights and Fundamental Freedoms ("the Convention") by a Bulgarian national, Ms Gana Petkova Velcheva ("the applicant"), on 30 June 2008.

2. The applicant was represented by Mr M. Ekimdzhiev and Ms G. Chernicherska, lawyers practising in Plovdiv. The Bulgarian Government ("the Government") were represented by their Agent, Ms Y. Stoyanova, of the Ministry of Justice.

3. The applicant alleged that the authorities had failed to comply with a final court judgment allowing her claim for restitution of agricultural land.

4. On 7 May 2013 the application was communicated to the Government.

**Figure 1** **Procedure.** This section contains the procedure followed before the Court, from the lodging of the individual application until the judgment was handed down.

*Submissions.* The second one comprises the arguments made by the Court itself on the *Merits*.

* **Parties' submissions:** The Parties' Submissions typically summarise the main arguments made by the applicant and the respondent state. Since in the vast majority of cases the material facts are taken for granted, having been authoritatively established by domestic courts, this part has almost exclusively to do with the legal arguments used by the parties.

* **Merits:** This subsection provides the legal reasons that purport to justify the specific outcome reached by the Court. Typically, the Court places its reasoning within a wider set of rules, principles and doctrines that have already been established in its past case-law and attempts to ground the decision by reference to these. It is to be expected, then, that this subsection refers almost exclusively to legal arguments, sometimes mingled with bits of factual information repeated from previous parts.

• **Operative provisions:** This is the section where the Court announces the outcome of the case, which is a decision to the effect that a violation of some Convention article either did or did not take place. Sometimes it is coupled with a decision on the division of legal costs and, much more rarely, with an indication of interim measures, under article 39 of the ECHR.

Figures 1–4, show extracts of different sections from the Case of "Velcheva v. Bulgaria" (http://hudoc.echr.coe.int/sites/eng/pages/search.aspx?i=001-155099) following the structure described above.

### Data

We create a data set[6] consisting of cases related to Articles 3, 6, and 8 of the Convention. We focus on these three articles for two main reasons. First, these articles provided the most data we could automatically scrape. Second, it is of crucial importance that there should be a sufficient number of cases available, in order to test the models. Cases from the selected articles fulfilled both criteria. Table 1 shows the Convention right that each article protects and the number of cases in our data set.

[6]The data set is publicly available for download from https://figshare.com/s/6f7d9e7c375ff0822564.

## THE FACTS

### I. THE CIRCUMSTANCES OF THE CASE

5. The applicant was born in 1927 and lives in the village of Ribaritsa.

6. Her father, of whom she is the sole heir, owned agricultural land in the area surrounding the village which was incorporated into an agricultural cooperative at the beginning of the 1950s.

7. In 1991, following the adoption of the Agricultural Land Act ("the ALA", see paragraph 17 below), the applicant applied for the land's restitution.

8. By a decision dated 10 March 1999 the land commission dealing with the case refused to restore her rights to two plots of 900 and 2,000 square metres respectively, noting that sheep pens had been built on them by the agricultural cooperative. It held that the applicant was entitled to compensation in lieu of restitution.

**Figure 2** **The facts.** This section comprises all material which is not considered as belonging to points of law, i.e., legal arguments.

### A. Arguments of the parties

#### 1. The Government

22. Referring to the Agriculture and Forestry Department's decision of 18 October 2006 (see paragraph 16 above) – of which the Court was not aware prior to communication of the present application – the Government argued that the applicant, in concealing its existence, had abused her right of individual application. On these grounds, the Government urged the Court to declare the application inadmissible.

23. On the merits, the Government argued that there had been no breach of the applicant's rights, because the judgment of 8 September 2005 had been enforced with the adoption of the decision of 18 October 2006. They contended that after this decision, and since the land claimed by the applicant had been transferred to a third party in 1995, it was up to the applicant to bring proceedings against that third party to defend her property rights.

**Figure 3** **The law.** The law section is focused on considering the merits of the case, through the use of legal argument.

## FOR THESE REASONS, THE COURT, UNANIMOUSLY,

1. *Declares* the application admissible;

2. *Holds* that there has been a violation of Article 6 § 1 of the Convention;

3. *Holds* that there has also been a violation of Article 1 of Protocol No. 1;

4. *Holds* that the question of the application of Article 41, insofar as it concerns the applicant's claims for pecuniary and non-pecuniary damage, is not ready for decision; accordingly,
   (a) reserves the said question;
   (b) invites the Government and the applicant to submit, within four months from the date on which the judgment becomes final in accordance with Article 44 § 2 of the Convention,

**Figure 4** **Operative provisions.** This is the section where the Court announces the outcome of the case, which is a decision to the effect that a violation of some Convention article either did or did not take place.

**Table 1  Articles of the Convention and number of cases in the data set.** Article numbers, Convention right that each article protects and the number of cases in our data set.

| Article | Human Right | Cases |
|---|---|---|
| 3 | Prohibits torture and inhuman and degrading treatment | 250 |
| 6 | Protects the right to a fair trial | 80 |
| 8 | Provides a right to respect for one's "private and family life, his home and his correspondence" | 254 |

For each article, we first retrieve all the cases available in HUDOC. Then, we keep only those that are in English and parse them following the case structure presented above. We then select an equal number of violation and non-violation cases for each particular article of the Convention. To achieve a balanced number of violation/non-violation cases, we first count the number of cases available in each class. Then, we choose all the cases in the smaller class and randomly select an equal number of cases from the larger class. This results to a total of 250, 80 and 254 cases for Articles 3, 6 and 8, respectively.

Finally, we extract the text under each part of the case by using regular expressions, making sure that any sections on operative provisions of the Court are excluded. In this way, we ensure that the models do not use information pertaining to the outcome of the case. We also preprocess the text by lower-casing and removing stop words (i.e., frequent words that do not carry significant semantic information) using the list provided by NLTK (https://raw.githubusercontent.com/nltk/nltk_data/ghpages/packages/corpora/stopwords.zip).

## Description of textual features

We derive textual features from the text extracted from each section (or subsection) of each case. These are either N-gram features, i.e., contiguous word sequences, or word clusters, i.e., abstract semantic topics.

- **N-gram features:** The Bag-of-Words (BOW) model (*Salton, Wong & Yang, 1975*; *Salton & McGill, 1986*) is a popular semantic representation of text used in NLP and Information Retrieval. In a BOW model, a document (or any text) is represented as the bag (multiset) of its words (unigrams) or N-grams without taking into account grammar, syntax and word order. That results to a vector space representation where documents are represented as $m$-dimensional variables over a set of $m$ N-grams. N-gram features have been shown to be effective in various supervised learning tasks (*Bamman, Eisenstein & Schnoebelen, 2014*; *Lampos & Cristianini, 2012*). For each set of cases in our data set, we compute the top-2000 most frequent N-grams where $N \in \{1, 2, 3, 4\}$. Each feature represents the normalized frequency of a particular N-gram in a case or a section of a case. This can be considered as a feature matrix, $C \in \mathbb{R}^{c \times m}$, where $c$ is the number of the cases and $m = 2,000$. We extract N-gram features for the Procedure (**Procedure**), Circumstances (**Circumstances**), Facts (**Facts**), Relevant Law (**Relevant Law**), Law (**Law**) and the Full case (**Full**) respectively. Note that the representations of the Facts is obtained by taking the mean vector of Circumstances and Relevant Law. In a similar

way, the representation of the Full case is computed by taking the mean vector of all of its sub-parts.

- **Topics:** We create topics for each article by clustering together N-grams that are semantically similar by leveraging the distributional hypothesis suggesting that similar words appear in similar contexts. We thus use the $C$ feature matrix (see above), which is a distributional representation (*Turney & Pantel, 2010*) of the N-grams given the case as the context; each column vector of the matrix represents an N-gram. Using this vector representation of words, we compute N-gram similarity using the cosine metric and create an N-gram by N-gram similarity matrix. We finally apply spectral clustering (*von Luxburg, 2007*)—which performs graph partitioning on the similarity matrix—to obtain 30 clusters of N-grams. For Articles 6 and 8, we use the Article 3 data for selecting the number of clusters $T$, where $T = \{10, 20, \ldots, 100\}$, while for Article 3 we use Article 8. Given that the obtained topics are hard clusters, an N-gram can only be part of a single topic. A representation of a cluster is derived by looking at the most frequent N-grams it contains. The main advantages of using topics (sets of N-grams) instead of single N-grams is that it reduces the dimensionality of the feature space, which is essential for feature selection, it limits overfitting to training data (*Lampos et al., 2014*; *Preoţiuc-Pietro, Lampos & Aletras, 2015*; *Preoţiuc-Pietro et al., 2015*) and also provides a more concise semantic representation.

## Classification model

The problem of predicting the decisions of the ECtHR is defined as a binary classification task. Our goal is to predict if, in the context of a particular case, there is a violation or non-violation in relation to a specific Article of the Convention. For that purpose, we use each set of textual features, i.e., N-grams and topics, to train Support Vector Machine (SVM) classifiers (*Vapnik, 1998*). An SVM is a machine learning algorithm that has shown particularly good results in text classification, especially using small data sets (*Joachims, 2002*; *Wang & Manning, 2012*). We employ a linear kernel since that allows us to identify important features that are indicative of each class by looking at the weight learned for each feature (*Chang & Lin, 2008*). We label all the violation cases as $+1$, while no violation is denoted by $-1$. Therefore, features assigned with positive weights are more indicative of violation, while features with negative weights are more indicative of no violation.

The models are trained and tested by applying a stratified 10-fold cross validation, which uses a held-out 10% of the data at each stage to measure predictive performance. The linear SVM has a regularisation parameter of the error term $C$, which is tuned using grid-search. For Articles 6 and 8, we use the Article 3 data for parameter tuning, while for Article 3 we use Article 8.

## RESULTS AND DISCUSSION

### Predictive accuracy

We compute the predictive performance of both sets of features on the classification of the ECtHR cases. Performance is computed as the mean accuracy obtained by 10-fold

**Table 2  Accuracy of the different feature types across articles.** Accuracy of predicting violation/non-violation of cases across articles on 10-fold cross-validation using an SVM with linear kernel. Parentheses contain the standard deviation from the mean. Accuracy of random guess is .50. **Bold** font denotes best accuracy in a particular Article or on Average across Articles.

| Feature Type | | Article 3 | Article 6 | Article 8 | Average |
|---|---|---|---|---|---|
| N-grams | Full | .70 (.10) | .82 (.11) | .72 (.05) | .75 |
| | Procedure | .67 (.09) | .81 (.13) | .71 (.06) | .73 |
| | Circumstances | .68 (.07) | .82 (.14) | .77 (.08) | .76 |
| | Relevant law | .68 (.13) | .78 (.08) | .72 (.11) | .73 |
| | Facts | .70 (.09) | .80 (.14) | .68 (.10) | .73 |
| | Law | .56 (.09) | .68 (.15) | .62 (.05) | .62 |
| Topics | | **.78 (.09)** | .81 (.12) | .76 (.09) | .78 |
| Topics and circumstances | | .75 (.10) | **.84 (0.11)** | **.78 (0.06)** | **.79** |

cross-validation. Accuracy is computed as follows:

$$\text{Accuracy} = \frac{TV + TNV}{V + NV} \tag{1}$$

where $TV$ and $TNV$ are the number of cases correctly classified that there is a violation an article of the Convention or not respectively. $V$ and $NV$ represent the total number of cases where there is a violation or not respectively.

Table 2 shows the accuracy of each set of features across articles using a linear SVM. The rightmost column also shows the mean accuracy across the three articles. In general, both N-gram and topic features achieve good predictive performance. Our main observation is that both language use and topicality are important factors that appear to stand as reliable proxies of judicial decisions. Therefore, we take a further look into the models by attempting to interpret the differences in accuracy.

We observe that 'Circumstances' is the best subsection to predict the decisions for cases in Articles 6 and 8, with a performance of .82 and .77 respectively. In Article 3, we obtain better predictive accuracy (.70) using the text extracted from the full case ('Full') while the performance of 'Circumstances' is almost comparable (.68). We should again note here that the 'Circumstances' subsection contains information regarding the factual background of the case, as this has been formulated by the Court. The subsection therefore refers to the actions and events which triggered the case and gave rise to a claim made by an individual to the effect that the ECHR was violated by some state. On the other hand, 'Full', which is a mixture of information contained in all of the sections of a case, surprisingly fails to improve over using only the 'Circumstances' subsection. This entails that the factual background contained in the 'Circumstances' is the most important textual part of the case when it comes to predicting the Court's decision.

The other sections and subsections that refer to the facts of a case, namely 'Procedure,' 'Relevant Law' and 'Facts' achieve somewhat lower performance (.73 cf. .76), although they remain consistently above chance. Recall, at this point, that the 'Procedure' subsection consists only of general details about the applicant, such as the applicant's name or country of origin and the procedure followed before domestic courts.

On the other hand, the 'Law' subsection, which refers either to the legal arguments used by the parties or to the legal reasons provided by the Court itself on the merits of a case consistently obtains the lowest performance (.62). One important reason for this poor performance is that a large number of cases does not include a 'Law' subsection, i.e., 162, 52 and 146 for Articles 3, 6 and 8 respectively. That happens in cases that the Court deems inadmissible, concluding to a judgment of non-violation.

We also observe that the predictive accuracy is high for all the Articles when using the 'Topics' as features, i.e., .78, .81 and .76 for Articles 3, 6 and 8 respectively. 'Topics' obtain the best performance in Article 3 and performance comparable to 'Circumstances' in Articles 6 and 8. 'Topics' form a more abstract way of representing the information contained in each case and capture a more general gist of the cases.

Combining the two best performing sets of features ('Circumstances' and 'Topics') we achieve the best average classification performance (.79). The combination also yields slightly better performance for Articles 6 and 8 while performance marginally drops for Article 3. That is .75, .84 and .78 for Articles 3, 6 and 8 respectively.

## Discussion

The consistently more robust predictive accuracy of the 'Circumstances' subsection suggests a strong correlation between the facts of a case, as these are formulated by the Court in this subsection, and the decisions made by judges. The relatively lower predictive accuracy of the 'Law' subsection could also be an indicator of the fact that legal reasons and arguments of a case have a weaker correlation with decisions made by the Court. However, this last remark should be seriously mitigated since, as we have already observed, many inadmissibility cases do not contain a separate 'Law' subsection.

### *Legal formalism and realism*

These results could be understood as providing some evidence for judicial decision-making approaches according to which judges are primarily responsive to non-legal, rather than to legal, reasons when they decide appellate cases. Without going into details with respect to a particularly complicated debate that is out of the scope of this paper, we may here simplify by observing that since the beginning of the 20th century, there has been a major contention between two opposing ways of making sense of judicial decision-making: legal formalism and legal realism (*Posner, 1986*; *Tamanaha, 2009*; *Leiter, 2010*). Very roughly, legal formalists have provided a *legal model* of judicial decision-making, claiming that the law is rationally determinate: judges either decide cases deductively, by subsuming facts under formal legal rules or use more complex legal reasoning than deduction whenever legal rules are insufficient to warrant a particular outcome (*Pound, 1908*; *Kennedy, 1973*; *Grey, 1983*; *Pildes, 1999*). On the other hand, legal realists have criticized formalist models, insisting that judges primarily decide appellate cases by responding to the stimulus of the facts of the case, rather than on the basis of legal rules or doctrine, which are in many occasions rationally indeterminate (*Llewellyn, 1996*; *Schauer, 1998*; *Baum, 2009*; *Leiter, 2007*; *Miles & Sunstein, 2008*).

Extensive empirical research on the decision-making processes of various supreme and international courts, and especially the US Supreme Court, has indicated rather consistently

that pure legal models, especially deductive ones, are false as an empirical matter when it comes to cases decided by courts further up the hierarchy. As a result, it is suggested that the best way to explain past decisions of such courts and to predict future ones is by placing emphasis on other kinds of empirical variables that affect judges (*Baum, 2009*; *Schauer, 1998*). For example, early legal realists had attempted to classify cases in terms of regularities that can help predict outcomes, in a way that did not reflect standard legal doctrine (*Llewellyn, 1996*). Likewise, the *attitudinal model* for the US Supreme Court claims that the best predictors of its decisions are the policy preferences of the Justices and not legal doctrinal arguments (*Segal & Spaeth, 2002*).

In general, and notwithstanding the simplified snapshot of a very complex debate that we just presented, our results could be understood as lending some support to the basic legal realist intuition according to which judges are primarily responsive to non-legal, rather than to legal, reasons when they decide hard cases. In particular, if we accept that the 'Circumstances' subsection, with all the caveats we have already voiced, is a (crude) proxy for non-legal facts and the 'Law' subsection is a (crude) proxy for legal reasons and arguments, the predictive superiority of the 'Circumstances' subsection seems to cohere with extant legal realist treatments of judicial decision-making.

However, not more should be read into this than our results allow. First, as we have already stressed at several occasions, the 'Circumstances' subsection is not a neutral statement of the facts of the case and we have only assumed the similarity of that subsection with analogous sections found in lodged applications and briefs. Second, it is important to underline that the results should also take into account the so-called *selection effect* (*Priest & Klein, 1984*) that pertains to cases judged by the ECtHR as an international court. Given that the largest percentage of applications never reaches the Chamber or, still less, the Grand Chamber, and that cases have already been tried at the national level, it could very well be the case that the set of ECtHR decisions on the merits primarily refers to cases in which the class of legal reasons, defined in a formal sense, is already considered as indeterminate by competent interpreters. This could help explain why judges primarily react to the facts of the case, rather than to legal arguments. Thus, further text-based analysis is needed in order to determine whether the results could generalise to other courts, especially to domestic courts deciding ECHR claims that are placed lower within the domestic judicial hierarchy. Third, our discussion of the realism/formalism debate is overtly simplified and does not imply that the results could not be interpreted in a sophisticated formalist way. Still, our work coheres well with a bulk of other empirical approaches in the legal realist vein.

### *Topic analysis*

The topics further exemplify this line of interpretation and provide proof of the usefulness of the NLP approach. The linear kernel of the SVM model can be used to examine which topics are most important for inferring whether an article of the Convention has been violated or not by looking at their weights $w$. Tables 3– 5 present the six topics for the most positive and negative SVM weights for the articles 3, 6 and 8, respectively. Topics identify in a sufficiently robust manner patterns of fact scenarios that correspond to well-established trends in the Court's case law.

**Table 3  The most predictive topics for Article 3 decisions.** Most predictive topics for Article 3, represented by the 20 most frequent words, listed in order of their SVM weight. Topic labels are manually added. Positive weights (*w*) denote more predictive topics for violation and negative weights for no violation.

| Topic | Label | Words | *w* |
|---|---|---|---|
| | | **Top-5 Violation** | |
| 4 | Positive State Obligations | injury, protection, ordered, damage, civil, caused, failed, claim, course, connection, region, effective, quashed, claimed, suffered, suspended, carry, compensation, pecuniary, ukraine | 13.50 |
| 10 | Detention conditions | prison, detainee, visit, well, regard, cpt, access, food, situation, problem, remained, living, support, visited, establishment, standard, admissibility merit, overcrowding, contact, good | 11.70 |
| 3 | Treatment by state officials | police, officer, treatment, police officer, July, ill, force, evidence, ill treatment, arrest, allegation, police station, subjected, arrested, brought, subsequently, allegedly, ten, treated, beaten | 10.20 |
| | | **Top-5 No Violation** | |
| 8 | Prior Violation of Article 2 | june, statement, three, dated, car, area, jurisdiction, gendarmerie, perpetrator, scene, June applicant, killing, prepared, bullet, wall, weapon, kidnapping, dated June, report dated, stopped | −12.40 |
| 19 | Issues of Proof | witness, asked, told, incident, brother, heard, submission, arrived, identity, hand, killed, called, involved, started, entered, find, policeman, returned, father, explained | −15.20 |
| 13 | Sentencing | sentence, year, life, circumstance, imprisonment, release, set, president, administration, sentenced, term, constitutional, federal, appealed, twenty, convicted, continued, regime, subject, responsible | −17.40 |

[7]Note that all the cases used as examples in this section are taken from the data set we used to perform the experiments.

First, topic 13 in Table 3 has to do with whether long prison sentences and other detention measures can amount to inhuman and degrading treatment under Article 3. That is correctly identified as typically not giving rise to a violation (*European Court of Human Rights, 2015*). For example, cases[7] such as Kafkaris v. Cyprus ([GC] no. 21906/04, ECHR 2008-I), Hutchinson v. UK (no. 57592/08 of 3 February 2015) and Enea v. Italy ([GC], no. 74912/01, ECHR 2009-IV) were identified as exemplifications of this trend. Likewise, topic 28 in Table 5 has to do with whether certain choices with regard to the social policy of states can amount to a violation of Article 8. That was correctly identified as typically not giving rise to a violation, in line with the Court's tendency to acknowledge a large margin of appreciation to states in this area (*Greer, 2000*). In this vein, cases such as Aune v. Norway (no. 52502/07 of 28 October 2010) and Ball v. Andorra (Application no. 40628/10 of 11 December 2012) are examples of cases where topic 28 is dominant. Similar observations apply, among other things, to topics 23, 24 and 27. That includes issues with the enforcement of domestic judgments giving rise to a violation of Article 6 (*Kiestra, 2014*). Some representative cases are Velskaya v. Russia, of 5 October 2006 and Aleksandrova v. Russia of 6 December 2007. Topic 7 in Table 4 is related to lower standard of review when property rights are at play (*Tsarapatsanis, 2015*). A representative

**Table 4 The most predictive topics for Article 6 decisions.** Most predictive topics for Article 6, represented by the 20 most frequent words, listed in order of their SVM weight. Topic labels are manually added. Positive weights ($w$) denote more predictive topics for violation and negative weights for no violation.

| Topic | Label | Words | $w$ |
|---|---|---|---|
| | | **Top-5 Violation** | |
| 27 | Enforcement of domestic judgments and reasonable time | appeal, enforcement, damage, instance, dismissed, established, brought, enforcement proceeding, execution, limit, court appeal, instance court, caused, time limit, individual, responsible, receipt, court decision, copy, employee | 11.70 |
| 23 | Enforcement of domestic judgments and reasonable time | court, applicant, article, judgment, case, law, proceeding, application, government, convention, time, article convention, January, human, lodged, domestic, February, September, relevant, represented | 9.15 |
| 24 | Enforcement of domestic judgments and reasonable time | party, final, respect, set, interest, alleged, general, violation, entitled, complained, obligation, read, fair, final judgment, violation article, served, applicant complained, summons, convention article, fine | 6.78 |
| | | **Top-5 No violation** | |
| 10 | Criminal limb | defendant, detention, witness, cell, counsel, condition, defence, court upheld, charged, serious, regional court upheld, pre, remand, inmate, pre trial, extended, detained, temporary, defence counsel, metre | −5.71 |
| 3 | Criminal limb | procedure, judge, fact, federal, justice, reason, charge, point, criminal procedure, code criminal, code criminal procedure, result, pursuant, article code, lay, procedural, point law, indictment, lay judge, argued, appeal point law | −7.01 |
| 7 | Property rights and claims by companies | compensation, company, property, examined, cassation, rejected, declared, owner, deputy, tula, returned, duly, enterprise, moscow, foreign, appears, control, violated, absence, transferred | −9.08 |

case here is Oao Plodovaya Kompaniya v. Russia of 7 June 2007. Consequently, the topics identify independently well-established trends in the case law without recourse to expert legal/doctrinal analysis.

The above observations require to be understood in a more mitigated way with respect to a (small) number of topics. For instance, most representative cases for topic 8 in Table 3 were not particularly informative. This is because these were cases involving a person's death, in which claims of violations of Article 3 (inhuman and degrading treatment) were only subsidiary: this means that the claims were mainly about Article 2, which protects the right to life. In these cases, the absence of a violation, even if correctly identified, is more of a technical issue on the part of the Court, which concentrates its attention on Article 2 and rarely, if ever, moves on to consider independently a violation of Article 3. This is exemplified by cases such as Buldan v. Turkey of 20 April 2004 and Nuray Şen v. Turkey of 30 March 2004, which were, again, correctly identified.

On the other hand, cases have been misclassified mainly because their textual information is similar to cases in the opposite class. We observed a number of cases where there is a

**Table 5** **The most predictive topics for Article 8 decisions.** Most predictive topics for Article 8, represented by the 20 most frequent words, listed in order of their SVM weight. Topic labels are manually added. Positive weights ($w$) denote more predictive topics for violation and negative weights for no violation.

| Topic | Label | Words | $w$ |
|---|---|---|---|
| | | **Top-5 Violation** | |
| 30 | Death and military action | son, body, result, russian, department, prosecutor office, death, group, relative, head, described, military, criminal investigation, burial, district prosecutor, men, deceased, town, attack, died | 15.70 |
| 1 | Unlawful limitation clauses | health moral, law democratic, law democratic society, disorder crime, prevention disorder, prevention disorder crime, economic well, protection health, interest national, interest national security, public authority exercise, interference public authority exercise, national security public, exercise law democratic, public authority exercise law, authority exercise law democratic, exercise law, authority exercise law, exercise law democratic society, crime protection | 12.20 |
| 26 | Judicial procedure | second, instance, second applicant, victim, municipal, violence, authorised, address, municipal court, relevant provision, behaviour, register, appear, maintenance, instance court, defence, procedural, decide, court decided, quashed | 9.51 |
| | | **Top-5 No violation** | |
| 25 | Discretion of state authorities | service, obligation, data, duty, review, high, system, test, concern, building, agreed, professional, positive, threat, carry, van, accepted, step, clear, panel | −7.89 |
| 28 | Social policy | contact, social, care, expert, opinion, living, welfare, county, physical, psychological, agreement, divorce, restriction, support, live, dismissed applicant, prior, remained, court considered, expressed | −12.30 |
| 4 | Migration cases | national, year, country, residence, minister, permit, requirement, netherlands, alien, board, claimed, stay, contrary, objection, spouse, residence permit, close, deputy, deportation, brother | −13.50 |

violation having a very similar feature vector to cases that there is no violation and vice versa.

## CONCLUSIONS

We presented the first systematic study on predicting judicial decisions of the European Court of Human Rights using only the textual information extracted from relevant sections of ECtHR judgments. We framed this task as a binary classification problem, where the training data consists of textual features extracted from given cases and the output is the actual decision made by the judges.

Apart from the strong predictive performance that our statistical NLP framework achieved, we have reported on a number of qualitative patterns that could potentially drive judicial decisions. More specifically, we observed that the information regarding the

factual background of the case as this is formulated by the Court in the relevant subsection of its judgments is the most important part obtaining on average the strongest predictive performance of the Court's decision outcome. We suggested that, even if understood only as a crude proxy and with all the caveats that we have highlighted, the rather robust correlation between the outcomes of cases and the text corresponding to fact patterns contained in the relevant subsections coheres well with other empirical work on judicial decision-making in hard cases and backs basic legal realist intuitions.

Finally, we believe that our study opens up avenues for future work, using different kinds of data (e.g., texts of individual applications, briefs submitted by parties or domestic judgments) coming from various sources (e.g., the European Court of Human Rights, national authorities, law firms). However, data access issues pose a significant barrier for scientists to work on such kinds of legal data. Large repositories like HUDOC, which are easily and freely accessible, are only case law databases. Access to other kinds of data, especially lodged applications and briefs, would enable further research in the intersection of legal science and artificial intelligence.

### Funding
DPP received funding from Templeton Religion Trust (https://www.templeton.org) grant number: TRT-0048. VL received funding from Engineering and Physical Sciences Research Council (http://www.epsrc.ac.uk) grant number: EP/K031953/1. The funders had no role in study design, data collection and analysis, decision to publish, or preparation of the manuscript.

### Grant Disclosures
The following grant information was disclosed by the authors:
Templeton Religion Trust: TRT-0048.
Engineering and Physical Sciences Research Council: EP/K031953/1.

### Competing Interests
Nikolaos Aletras is an employee of Amazon.com, Cambridge, UK, but work was completed while at University College London.

### Author Contributions
- Nikolaos Aletras and Vasileios Lampos conceived and designed the experiments, performed the experiments, analyzed the data, contributed reagents/materials/analysis tools, wrote the paper, prepared figures and/or tables, performed the computation work, reviewed drafts of the paper.
- Dimitrios Tsarapatsanis conceived and designed the experiments, analyzed the data, contributed reagents/materials/analysis tools, wrote the paper, prepared figures and/or tables, reviewed drafts of the paper.

- Daniel Preoţiuc-Pietro conceived and designed the experiments, contributed reagents/materials/analysis tools, wrote the paper, prepared figures and/or tables, reviewed drafts of the paper.

### Data Availability

ECHR dataset: https://figshare.com/s/6f7d9e7c375ff0822564.

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
