# Peer review of "Predicting judicial decisions of the European Court of Human Rights: a Natural Language Processing perspective"

_PeerJ Computer Science, doi:10.7717/peerj-cs.93_

## Round 0.1 · original submission · Major Revisions

I very much like the topic of this paper. Both reviewers saw the value of this manuscript although each raised concerns about the reporting, experiments, and interpretation of results.

I strongly encourage the authors to address the concerns of reviewer 1, and carefully quantify the results and its interpretations in a legal context. I also strongly encourage the authors to take into account the suggestions of reviewer 2 in clarifying the content, discussing failure cases so as to get insight, and having not just two features separately, but also consider combining them.

I look forward to the revised manuscript.

Reviewer 1 ·

Basic reporting

No comments.

Experimental design

This review is prepared by a referee without training in quantitative methods, in computer science, or in natural language processing, and as such should be read with an appropriately-sized grain of salt.
The description, under 'Data', of how cases were selected needs clarification in several ways (lines 165-182).
First, when it is said that Article 3 'Prohibits torture', are we to understand that the study does not cover the other prohibitions contained in Article 3 (such as the prohibition on inhuman treatment)? Precision is important in legal writing and it is important here.
Second, the number of cases seems much lower than what one would expect, and much lower than what a rudimentary search of the HUDOC database generates. For example, a basic HUDOC search of Article 6, in English, generates over 10,000 cases. Of those, HUDOC indicates there at least 8,000 violation cases and at least 900 non-violation cases. If the methodology at lines 172-177 is replicable, one would expect that this study would have included at least 1,800 cases in its study of Article 6 (being all the cases in the smaller class plus a randomly selected equal number of cases from the larger class). And yet here the number studied is just 80. It is hard to discern the reasons for the discrepancy between the draft article and the results of a basic HUDOC search.
Third, the reasons for choosing articles 3, 6, and 8 could be substantiated a bit more. Surely *all* of the ECHR rights may be regarded as "important human rights that correspond to a variety of interests" (lines 167-168)? Why focus on these three?

Validity of the findings

This review is prepared by a referee without training in quantitative methods, in computer science, or in natural language processing, and as such should be read with an appropriately-sized grain of salt.
The article claims that "there is a strong correlation between the actual facts of a case and the decisions made by judges" (lines 264-265). I have serious concerns about whether or not this conclusion is substantiated by the data which preceded the Discussion. First, it is unclear what the authors mean by "the *actual* facts" (line 265). Are the "actual" facts somehow different from "the facts"? Second, it is not clear what the authors mean when they say "information available to the judges before they make any comments or decisions" (line 180). Are the authors implying that the judgments contain all the information available to the judges before they made their decision? If so, this would seem to be a misunderstanding of how courts work (surely, at the very least, one would want to look at the full written arguments of the parties, rather than simply the summaries of those written arguments that are contained in judgments?). Third, it seems to me that this study proves, at best, that there is a correlation between the facts *as described in the judgement* and the result of a case. There is a difference between "the actual facts of a case" and "the facts as they are described in the judgment of case". The article does not acknowledge this difference at all. This is a problem. I'm afraid that the authors seem to be under the impression that the facts section of a judgment is an objective scientifically-established recitation of the facts. Unless the authors are aware of ECtHR practice that I am unaware of, this seems dangerously naive. On my understanding, the judgments of the ECtHR are prepared by the judges, their assistants, and the Court Registry. In any court anywhere around the world, including the ECtHR, it would not be unusual in the slightest for the judges, the assistants, or the registry, to frame the facts in light of their full understanding of the case (which would include their view on whether or not there is a violation). Facts sections of judgments are not peer-reviewed scientific papers. They are subjective summaries of the facts, including what the authors think is relevant and what they think is irrelevant. If a judge/judicial assistant/registrar is of the preliminary view that a violation is likely, it would not be at all unusual for them to frame the facts differently than how those same facts would be framed if they were of the view that a violation was unlikely. This raises a problem: the article involves the authors taking the facts section of a delivered judgment, and then predicting whether or not that same judgment will result in a violation or not. This may be useful, but it does not seem to be the same as "predicting judicial decisions...using only the textual information available to the judges before they make any comments or decisions about a specific case..." (lines 312-313). The model does not seem to provide any capability for ex ante prediction -- i.e. it does not allow the result of a judicial decision to be predicted until the facts section of the judgment can be analysed (and the facts section of the judgment cannot be analysed until the judgment is handed down). Surely this limits its utility?
Perhaps I misunderstand the mathematical value of the study; perhaps I misunderstand the internal workings of the European Court. But even if I am wrong on the maths or on the workings of the Court, the article needs to be considerably clearer about what it is predicting and about the nature of how judgments are written and prepared. Without that, it is hard to attach too much significance to its findings, I'm afraid.
Fourth, the authors claim that their study amounts to support for legal realism over legal formalism (line 322). This may be so, but a much more sophisticated account (than that at lines 28-37) of the debate about realism and formalism would be needed to draw much of a conclusion here.

·

Basic reporting

This paper is about applying the state-of-the-art natural language processing analysis and a machine learning algorithm to build a binary classifier to predict judicial decisions.

The paper is clearly written and easy to understand. The Authors have followed the appropriate document structure and the submission is self-contained.
The authors had made the data set annotation public-ally available, and the content of the cases can be download from the European Court of Human Rights (EctHR) website.

Experimental design

The research question is clearly defined: it is possible to use text processing and machine learning to predict whether, given a case, there has been a violation of an article in the convention of human rights. The research question is certainly relevant, and the results are interesting for the natural language processing and machine learning community, but it’s unclear how these findings are useful for the law and human rights community, since nothing is mentioned in the paper with regards to this question, for example, 'it would be useful to apply this kind of classifier as a tagging tool for highlighting cases in which violation of human rights are likely to be true? perhaps as a prioritizing or filtering means?'
This is more a curiosity than a negative comment.

From the natural language processing and machine learning point of view, the methodology is correct. The authors use well known features as bag-of-words and topic models as well as an appropriate classifier (SVM).
Nevertheless, there are a few weak points that are not explained or addressed in the paper.
- Topic models: in spectral clustering, the number of topics is input parameter, but nothing is mentioned about how the value of this parameter, in this case 30 topics, was chosen.
- I was expecting to see experiments using both set of features, bow and topics, but results are only reported for experiments using one set of features at a time. Why are there no experiments that combine both features sets? If the performance was lower than when using the individual features sets, the outcome is still useful for the community and should be reported.

Validity of the findings

Accuracy is reported as the evaluation metric used to measure performance, but nothing is said about how accuracy is calculated, the formula or an explanation would be helpful. I'm assuming accuracy should be understood to mean: the proportion of true outcomes (true positives and true negatives) among the total number of cases. Please, clarify this.

In the discussion section, the authors gave examples how topics aligned with the theme of some of the cases, but it’s hard to understand if those examples are from the dataset used or from another dataset. I figured it out by exploring the dataset itself. A line clarifying this would be helpful.

Comment something about the cases that were wrongly classify, for example, is there any commonality between the wrongly classified instances? What does it mean for the law community to have more than 20% of its cases wrongly classified?

Additional comments

The topic of the paper is very interesting and the paper is easy to follow, but I would like to read about how this results can be use by the law community.

---

## Round 0.2 · Minor Revisions

I'd like to commend the authors for carefully taking into account the reviewer comments and revising the manuscript.

The remaining concerns are still in regard to paying attention to the nuances (from both the legal and CS perspectives) of the claims being made, and to paint a realistic picture about how the proposed analysis could be used, or developed into something that the legal practice can use.

The availability of data is, and will be an ongoing discussion within the law and CS communities. This paper is well-positioned to lead the academic and legal practice community in this discussion/debate about what is the best way to share data that allows the technology to move forward.

I strongly encourage the authors to take into account these comments and send in another revision.

Reviewer 1 ·

Basic reporting

No comments.

Experimental design

I have been asked to re-consider a revised draft article entitled ‘Predicting Judicial Decisions of the European Court of Human Rights: A Natural Language Processing Perspective’ for ‘PeerJ CS’.

As with the earlier review, this review is prepared by a referee without training in quantitative methods, in computer science, or in natural language processing, and as such should be read with an appropriately-sized grain of salt.

The authors have responded to feedback thoughtfully, which must be commended. But I am afraid that, subject to the caveats above, the article’s legal analysis remains problematic. I don't wish to be unkind or unfair with any of these comments: I look forward to a day when this sort of model can succeed. But this paper needs further work.

(all line references are to the track-changes pdf version)

At lines 88-94, the authors suggest the model can ‘be used to rapidly identify cases’, perhaps in ‘submitted cases’. It remains fundamentally unclear to me what text the authors imagine the model will be operating on to make ex ante predictions, given that it works solely on text in completed judgments. Is it not at least worth *identifying* what the text is that the model would operate on in order to make its ‘rapid’ ex ante predictions? Otherwise, the risk is that the argument will appear to be limited to this: ‘Our model allows us to analyse how the facts are summarized in published cases and predict how, later in that very same published case, the case will be resolved’. This may be interesting in itself, but I’m not sure it’s what the authors want the model to do.

The new section at lines 140-166 seems problematic, unless I have misunderstood key aspects of it.
*First, the authors say that the ECtHR has very limited fact-finding powers (true). But the authors then move on, without citing any authority, to say that this means that ‘in the vast majority of cases, [the ECtHR] will defer…to the judgments of domestic courts that have already heard and dismissed the applicants’ complaint’. This is problematic in various ways (not least that it implies that the domestic courts hear and dismiss complaints on the same legal questions that the ECtHR does, which seems to suggest the ECtHR is an appellate court – more on this below). More problematically, the authors’ logic is unclear: why wouldn’t the ECtHR defer to the summary of the facts prepared by (e.g.) the government lawyers? Moreover, even if it were true that the Court defers in this manner ‘in the vast majority of cases’, surely it should not be difficult to find a range of law journal articles and analysis supporting this legal or procedural proposition? And finally, this logic would suggest that any predictive model should look principally or exclusively at the domestic courts' factual summaries, would it not?
*Second, the authors say that ‘the Court cannot openly acknowledge any kind of bias on its part’ and therefore ‘on their face, summaries of facts…have to be at least framed in as neutral…’. The significance of this point is unexplained. Are the authors arguing that the Court does, in reality, prepare neutral summaries? Or that it may well be biased but that it must hide that bias? How does this help the argument about making ex ante predictions of any sort? Moreover, the authors do not seem to appreciate here that outright bias is only one problem: the bigger problem for their argument is the possibility of perfectly rational differential emphasis by the judges/registry/etc of facts that they know will be significant *because they are also involved in reaching the legal conclusions*. Unless I have misunderstood the ECtHR's procedure, in which case I apologize.
*Third, the absence of disputes before the ECtHR about the facts does not mean that there cannot be different facts emphasized or prioritized by the judges/registry in light of the analysis that they most likely know will follow.
*Fourth, the authors say ‘the “Circumstances” subsection is the closest (even if sometimes crude) proxy we have to a reliable textual representative of the factual background of a case’. Perhaps this is so, but one may wonder about what ‘reliable’ is worth here without a clearer sense of the model's utility. Surely the facts as summarized in government and/or applicant arguments might be worth a look if the goal is to assist the Court/lawyers in making ex ante predictions about how cases will be decided, even if they are not 'reliable' in the way a peer-reviewed paper might be reliable?

At several points (line 325, line 340), the ECtHR seems to be referred to as an appellate court. It is not an appellate court. This means, (1) the range of orders and remedies available to the ECtHR are not those of an appellate court, with consequences for its analysis; (2) the ECtHR will frequently be applying different *legal* tests to those applied by the domestic appellate courts (eg, ‘was there a violation of Art5 or Art6?’ vs ‘was the conviction unsafe?’), and (3) therefore different *facts* or different emphasis on those facts may be of interest to the ECtHR than those that were of interest to the domestic court.

Validity of the findings

See above.

Additional comments

Please let me add: this is really creative and interesting work and I hope that it will be able to be developed to a point where it is of utility to scholars and lawyers in all disciplines. Perhaps I misunderstand the mathematical value of the study; perhaps I misunderstand the internal workings of the ECtHR; but at present I am afraid I simply do not think the legal analysis is sufficient.

·

Basic reporting

The authors of the papers have answered the reviewers comments in a satisfactory manner.
Firstly, and more important, relevant concerns from reviewer 1, who have a strong background on Law were addressed, especially in nuancing the conclusion about the correlation between the Facts and the Court decision and using a more precise legal writing along the paper.
Also, clarification about the Facts section and its subsection ‘Circumstances of the case’ make the paper more convincingly and easy to follow.
From the technical part, clarification on the evaluation metric used and further experimentation with combined features sets were carry out, making the paper more substantial and methodologically stronger, and improving the overall accuracy of the predictor.

The authors commented during the review process that there are some barriers for accessing the data, from the EctHR portal. Besides, the authors mentioned that accessing cases from domestic courts is not straightforward either.
I would like to see a comment regarding data access issues, perhaps a sentence or two in the conclusion section. Are data access issues stopping or slowing down emerging research as the one presented in this paper? Should cases in the EctHR and domestic cases be easily and freely available? If that it the case, is the EctHR making progress in open their repositories for public good?

Looking forward to see this paper published in the journal.

Experimental design

No-comments

Validity of the findings

No-comments

Additional comments

No-comments

---

## Author Rebuttal · Round 0.2

We would like to thank the editor and the reviewers for their constructive comments. See below our point-by-point response.

**Reviewer 1**

**1. The description, under 'Data', of how cases were selected needs clarification in several ways (lines 165-182).**
**First, when it is said that Article 3 'Prohibits torture', are we to understand that the study does not cover the other prohibitions contained in Article 3 (such as the prohibition on inhuman treatment)? Precision is important in legal writing and it is important here.**

The study does cover the other prohibitions contained in Article 3 (such as inhuman and degrading treatment). Accordingly, we have added the full description of the Articles in Table 1.

**2. Second, the number of cases seems much lower than what one would expect, and much lower than what a rudimentary search of the HUDOC database generates. For example, a basic HUDOC search of Article 6, in English, generates over 10,000 cases. Of those, HUDOC indicates there at least 8,000 violation cases and at least 900 non-violation cases. If the methodology at lines 172-177 is replicable, one would expect that this study would have included at least 1,800 cases in its study of Article 6 (being all the cases in the smaller class plus a randomly selected equal number of cases from the larger class). And yet here the number studied is just 80. It is hard to discern the reasons for the discrepancy between the draft article and the results of a basic HUDOC search.**

At the beginning of our study, we planned to manually develop the data set using experts in the School of Law of the University of Sheffield. We quickly realised that this process was very slow and thus infeasible. Then, we decided to automate the process by devising a "reasonable" common structure in the format of the case reports. This includes the main parts of "Procedure", "The Facts", "The Law" and "Operative Provisions" in that order. We strictly filtered out cases that failed to match more than one of these main sections. There are many cases that a different wording is used in the title of a section which makes it difficult to be captured automatically. In addition, we strictly filtered out comments made by the Court keeping only those comments made by the two parties. That sometimes resulted into empty sections. Finally, many case reports retrieved were actually in French even if the selected language was set to English. For these reasons, our dataset contains a smaller number of cases. However, the results obtained are significantly different compared to the random baseline, i.e. 50% accuracy (t-test, $p<0.001$). Note that we scraped the Hudoc website and matched cases with regular expressions without having access to the actual database (access here means to be able to retrieve data automatically using some sort of database query language such as SQL and not through the website interface). We believe that our results can be seen as a proof of concept and by given access to the actual database we could perform a study that covers all the available cases and articles. That would be an interesting avenue for future work and/or a research grant proposal.

**3. Third, the reasons for choosing articles 3, 6, and 8 could be substantiated a bit more. Surely *all* of the ECHR rights may be regarded as "important human rights that correspond to a variety of interests" (lines 167-168)? Why focus on these three?**

These Articles seemed to us to provide the most data we could automatically scrape. We have revised the text accordingly.

**4. The article claims that "there is a strong correlation between the actual facts of a case and the decisions made by judges" (lines 264-265). I have serious concerns about whether or not this conclusion is substantiated by the data which preceded the Discussion. First, it is unclear what the authors mean by "the *actual* facts" (line 265). Are the "actual" facts somehow different from "the facts"?**

No, they are not. We have revised the text accordingly.

**5. Second, it is not clear what the authors mean when they say "information available to the judges before they make any comments or decisions" (line 180). Are the authors implying that the judgments contain all the information available to the judges before they made their decision? If so, this would seem to be a misunderstanding of how courts work (surely, at the very least, one would want to look at the full written arguments of the parties, rather than simply the summaries of those written arguments that are contained in judgments?).**

Our wording was somewhat sloppy. All we meant to say was that the models do not have information pertaining to the operative provisions. We have revised the text accordingly.

**6. Third, it seems to me that this study proves, at best, that there is a correlation between the facts *as described in the judgement* and the result of a case. There is a difference between "the actual facts of a case" and "the facts as they are described in the judgment of case". The article does not acknowledge this difference at all. This is a problem. I'm afraid that the authors seem to be under the impression that the facts section of a judgment is an objective scientifically-established recitation of the facts. Unless the authors are aware of ECtHR practice that I am unaware of, this seems dangerously naive. On my understanding, the judgments of the ECtHR are prepared by the judges, their assistants, and the Court Registry. In any court anywhere around the world, including the ECtHR, it would not be unusual in the slightest for the judges, the assistants, or the registry, to frame the facts in light of their full understanding of the case (which would include their view on whether or not there is a violation). Facts sections of judgments are not peer-reviewed scientific papers. They are subjective summaries of the facts, including what the authors think is relevant and what they think is irrelevant. If a judge/judicial assistant/registrar is of the preliminary view that**

**a violation is likely, it would not be at all unusual for them to frame the facts differently than how those same facts would be framed if they were of the view that a violation was unlikely. This raises a problem: the article involves the authors taking the facts section of a delivered judgment, and then predicting whether or not that same judgment will result in a violation or not. This may be useful, but it does not seem to be the same as "predicting judicial decisions...using only the textual information available to the judges before they make any comments or decisions about a specific case..." (lines 312-313). The model does not seem to provide any capability for ex ante prediction -- i.e. it does not allow the result of a judicial decision to be predicted until the facts section of the judgment can be analysed (and the facts section of the judgment cannot be analysed until the judgment is handed down). Surely this limits its utility? Perhaps I misunderstand the mathematical value of the study; perhaps I misunderstand the internal workings of the European Court. But even if I am wrong on the maths or on the workings of the Court, the article needs to be considerably clearer about what it is predicting and about the nature of how judgments are written and prepared. Without that, it is hard to attach too much significance to its findings, I'm afraid.**

We should have made our argument clearer here. We made revisions to the text accordingly, stressing the following points. First, the ECtHR has only limited fact-finding powers, which implies that, in the vast majority of cases, it will defer, when summarizing the facts, to the judgments of domestic courts that have already heard and dismissed the applicants' complaint. While these can also reflect assumptions about relevance, they also reflect understandings of the facts that have been validated by more than one decision-maker. Second, the Court cannot openly acknowledge any kind of bias on its part. This means that, on their face, summaries of facts have to be at least framed in as neutral a way as possible. Furthermore, a random reading of ECtHR cases indicates that, in the vast majority of cases, parties do not seem to dispute the facts themselves, but merely their legal significance (i.e. whether a violation took place or not, given those facts). Third, it is important to note that the data used by the model are to do with 'the facts of the case are these are described in the relevant section of the judgment', as the reviewer correctly suggested. We have revised all pertinent formulations accordingly. Fourth, for our argument to get off the ground all we need is that the text of this section performs differently from the text of other sections. This much has been established by our model. Fifth, the reviewer is right that we should deflate our claim: the model is only a (crude) proxy to different kinds of considerations, and not a perfect representative of these considerations. We have revised all pertinent formulations accordingly. Sixth, this of course leaves open the possibility, noted by the reviewer, that this section is indeed formulated in a way that reflects judges' understanding of the case, which includes various judgments relating to relevance/irrelevance and, potentially, to biases related to how the case should be decided. We have acknowledged this openly, revising all pertinent formulations. Seventh, and final, point: insofar as the 'facts' section of the case is a (crude) proxy, it is an open question whether it could provide a basis for ex ante predictions of judgments. We do not really see any reason why it could not, since it — at the very least — proves the concept that, on the basis of chunks of particular textual information that differ on their face, it can do a relatively good job at predicting outcomes. So the model could have practical utility in this respect.

**7. Fourth, the authors claim that their study amounts to support for legal realism over legal formalism (line 322). This may be so, but a much more sophisticated account (than that at lines 28-37) of the debate about realism and formalism would be needed to draw much of a conclusion here.**

We have (a) moved the relevant points from the introduction to the discussion part and (b) deflated our claims accordingly, with all the usual caveats. Different (sub)sections of a judgment are not to be understood as more than crude proxies (but they are all that we have at this point). Again, we believe that the important thing is that the model, given the data, differentiates clearly between (sub)sections of a judgment and that is a significant result in itself.

**Reviewer 2**

**8. The research question is clearly defined: it is possible to use text processing and machine learning to predict whether, given a case, there has been a violation of an article in the convention of human rights. The research question is certainly relevant, and the results are interesting for the natural language processing and machine learning community, but it's unclear how these findings are useful for the law and human rights community, since nothing is mentioned in the paper with regards to this question, for example, 'it would be useful to apply this kind of classifier as a tagging tool for highlighting cases in which violation of human rights are likely to be true? perhaps as a prioritizing or filtering means?'**

From a more "applied" perspective, we mention in the abstract that "This can be useful, for both lawyers and judges, as an assisting tool to rapidly identify cases and extract patterns which lead to certain decisions.". In the revised version we have added that in the introduction as well. Moreover, insofar as different sections of the judgment can be understood as (crude) proxies of the relevance of different kinds of considerations to judicial decision-making, the analysis provides a first step that could be later further tested with text coming from lawyers' briefs/applications or domestic judgments. The hard part is to have access to that data (and that is why we focused on the ECtHR's judgments).

**9. Topic models: in spectral clustering, the number of topics is input parameter, but nothing is mentioned about how the value of this parameter, in this case 30 topics, was chosen.**

We tuned this parameter using the same strategy followed to tune the SVM parameters. We have added the description of how we set the value of this parameter in the Classification Model subsection.

**10. I was expecting to see experiments using both set of features, bow and topics, but results are only reported for experiments using one set of features at a time. Why are there no experiments that combine both features sets? If the performance was lower than when using the individual features sets, the outcome is still useful for the community and should be reported.**

We performed experiments combining both sets of features (N-grams Circumstances + Topics) yielding slightly better performance for articles 6 and 8 while performance was slightly lower for article 3. That is 0.75 (0.10), 0.84 (0.11) and 0.78 (0.06). We have updated Table 2 accordingly.

**11. Accuracy is reported as the evaluation metric used to measure performance, but nothing is said about how accuracy is calculated, the formula or an explanation would be helpful. I'm assuming accuracy should be understood to mean: the proportion of**

**true outcomes (true positives and true negatives) among the total number of cases. Please, clarify this.**

That is true. We have added the equation in section "Results and Discussion".

**12. In the discussion section, the authors gave examples how topics aligned with the theme of some of the cases, but it's hard to understand if those examples are from the dataset used or from another dataset. I figured it out by exploring the dataset itself. A line clarifying this would be helpful.**

The examples of cases we use in the discussion are from the dataset used in the study. We clarify that in the revised version of the paper.

**13. Comment something about the cases that were wrongly classify, for example, is there any commonality between the wrongly classified instances? What does it mean for the law community to have more than 20% of its cases wrongly classified?**

On the other hand, cases have been misclassified mainly because their textual information is similar to cases in the opposite class. We observed a number of cases where there is a violation having a very similar feature vector to cases that there is no violation and vice versa. We have added that comment in the Discussion subsection (Results section).

**14. The topic of the paper is very interesting and the paper is easy to follow, but I would like to read about how this results can be use by the law community.**

See point 8.

---

## Round 0.3 · Minor Revisions

The editor appreciates the author's continued effort to improve the manuscript -- and the reviewers detailed speedy feedback.

I recommend revision of the wording as the reviewer suggests, and look forward to see a manuscript of great readability for both the computer scientists and the law community.

Reviewer 1 ·

Basic reporting

Fine.

Experimental design

Fine, subject to the below.

Validity of the findings

Fine, subject to the below.

Additional comments

The caveats in the rebuttal and the changes made to the paper are interesting. The relevant limitations that are acknowledged -- those that make the paper reliant on crude proxies -- mean that the paper's arguments would struggle to be satisfactory in a legal paper or in legal argument. I understand that the methods adopted in this area may be different.

Four specific revisions:
(1) The the rule on exhaustion of domestic remedies seems to be poorly articulated, or perhaps misunderstood, at line 152. In particular, its application at the domestic level seems back-to-front, at least as it is explained here. It should be revised.
(2) If there is literature to support the assumption made in lines 108-111, that literature should be noted. Or is it the literature in lines 134-173?
(3) The authors have explained eloquently the limitations on their methodology due to unavailable data. It is therefore jarring to see an expression of belief at line 116-117 ("we believe there is") that, surely, is unsupported due to those very limitations? Is belief the basis for scientific argument here?
(4) Line 416 says "Large repositories like HUDOC should be easily and freely accessible." This could be more precise: is HUDOC not easily and freely accessible? Isn't the authors' real concern with the fact that HUDOC is a "case law database" and not a "database of case law and other things"?

As I have said before, this is really creative and interesting work and I hope that it will be able to be developed to a point where it is of utility to scholars in all disciplines and to lawyers.

---

## Author Rebuttal · Round 0.3

We would like to thank the editor and the reviewers for their constructive comments. See below our point-by-point response.

**Reviewer 1**

**1. At lines 88-94, the authors suggest the model can 'be used to rapidly identify cases', perhaps in 'submitted cases'. It remains fundamentally unclear to me what text the authors imagine the model will be operating on to make ex ante predictions, given that it works solely on text in completed judgments. Is it not at least worth \*identifying\* what the text is that the model would operate on in order to make its 'rapid' ex ante predictions? Otherwise, the risk is that the argument will appear to be limited to this: 'Our model allows us to analyse how the facts are summarized in published cases and predict how, later in that very same published case, the case will be resolved'. This may be interesting in itself, but I'm not sure it's what the authors want the model to do.**

Our main argument in favour of ex ante predictions of outcomes rests on the premise that there is enough similarity between (at least certain) chunks of text of completed judgments by the Court and other kinds of textual materials, to wit: (a) applications lodged with the Court, (b) briefs submitted by parties with respect to pending cases making ECHR-related legal arguments and, possibly, (c) sections of domestic judgments that touch upon ECHR-related issues (whether an application to the Court has been made or not). Since our predictive NLP approach (and not the specific supervised model *per se*) seems to work reasonably well with text chucks from published judgments issued by the Court, it could also be further tested to assess whether it in fact can generalise to these other kinds of (in our view quite similar) texts. Unfortunately, this is something that we cannot test at the moment, because we do not have access to the data set that will allow us to do so (and we will not have access to such material in the foreseeable future, since the Court does not easily give access to lodged applications or briefs submitted by parties, which would be the most natural candidates). We thus used published judgments as proxies for the material to which we do not have access. At the very least, our work proves the following point (and is therefore at least a 'proof of concept' in this limited sense): *if* there is enough similarity between the chunks of text that we analysed and/or applications and briefs, as we believe there is, *then* NLP approaches can be fruitfully used to predict outcomes with a certain degree of reliability (at the very least, with a degree of reliability that appears to exceed by far the random 50% distribution). So, at the very least, our work provides some justification for further research, potentially on a wider scale and with the use of materials to which we do not at the present moment have access. We hasten to add that this kind of research programme requires different kinds of resources than the ones currently at our disposal, since it renders imperative a close cooperation with the Court itself, with all that such a cooperation entails. Moreover, we provided various reasons to lend support to our view according to which the sections of the cases we analysed can *indeed* be used as proxies for these other sorts of materials (i.e. applications and briefs), and we shall get back to these reasons in the remaining sections of our answer. Overall, we do not believe that a wholesale scepticism with regard to such uses of sections of cases is warranted. We have incorporated the above clarifications in the article.

**2. The new section at lines 140-166 seems problematic, unless I have misunderstood key aspects of it. First, the authors say that the ECtHR has very limited fact-finding powers (true). But the authors then move on, without citing any authority, to say that this means that 'in the vast majority of cases, [the ECtHR] will defer…to the judgments of domestic courts that have already heard and dismissed the applicants' complaint'. This is problematic in various ways (not least that it implies that the domestic courts hear and dismiss complaints on the same legal questions that the ECtHR does, which seems to suggest the ECtHR is an appellate court – more on this below). More problematically, the authors' logic is unclear: why wouldn't the ECtHR defer to the summary of the facts prepared by (e.g.) the government lawyers? Moreover, even if it were true that the Court defers in this manner 'in the vast majority of cases', surely it should not be difficult to find a range of law journal articles and analysis supporting this legal or procedural proposition? And finally, this logic would suggest that any predictive model should look principally or exclusively at the domestic courts' factual summaries, would it not?**

There are various things to note here. First, we explicitly say that deference to domestic judgments is to do *only with summaries of the factual background of the case* and not with anything else. The exact phrase we had used is this "in the vast majority of cases, it [the ECtHR] will defer, *when summarizing the factual background of a case*, to the judgments of domestic courts" (emphasis added). As a result, we nowhere either said explicitly or implied that domestic courts hear and dismiss complaints on the same legal questions as the ECtHR does. In any event, though, we have taken note of the reviewer's comments and have revised the text accordingly, to remove any possibility of misunderstanding. Second, we are of the view that the main thing to underline is that the facts of the case, in the vast majority of judgments where the Court does not use its powers of investigation, are *not in dispute by the parties*. Accordingly, for the vast majority of cases, the content of those facts has been fixed by the rules of procedure (and evidence) used at the domestic level by national jurisdictions. Moreover, in the vast majority of cases where the facts are thus fixed, the main question that the Court has to answer is a legal (whether there has been a violation of some ECHR-protected right) and not a factual (what are the facts of this particular case) one. We thus believe that it can be reasonably held that, in view of the above, the question of the exact source used to summarize the facts is of secondary importance. We also wish to stress that 'reasonably' in the above sense does not mean 'bulletproof' for all intents and purposes. As a matter of empirical fact, we can never be sure about the exact ways in which the Court arrives at summaries of the facts of each case and this is a limitation to our analysis that we have revised our text to take account of. In addition, we have revised the text of our article to provide references to academic work on how registry lawyers prepare summaries of the facts for judges (an issue on which some -limited- socio-legal research has been conducted). Again, our argument here depends on certain (in our view) reasonable assumptions. We cannot prove these assumptions, since we do not have access to the materials themselves, so readers are invited to read our argument as a hypothetical: *if* these assumptions are met (which we think they do), *then* certain consequences follow. Third, In order to back our claim that the Court mostly defers to domestic courts when determining the

factual background of a case, we provided one reference to the leading research on the matter and one more reference to a textbook by a leading legal practitioner of the ECHR.

**3. Second, the authors say that 'the Court cannot openly acknowledge any kind of bias on its part' and therefore 'on their face, summaries of facts…have to be at least framed in as neutral…'. The significance of this point is unexplained. Are the authors arguing that the Court does, in reality, prepare neutral summaries? Or that it may well be biased but that it must hide that bias? How does this help the argument about making ex ante predictions of any sort? Moreover, the authors do not seem to appreciate here that outright bias is only one problem: the bigger problem for their argument is the possibility of perfectly rational differential emphasis by the judges/registry/etc of facts that they know will be significant \*because they are also involved in reaching the legal conclusions\*. Unless I have misunderstood the ECtHR's procedure, in which case I apologize.**

As we have already mentioned, we are not in a position to either corroborate or refute empirically the proposition to the effect that the fact summaries prepared by the Court are neutral or not. The point we tried to make is merely that, since the Court has to (at least) appear unbiased to both parties in a dispute, it has an interest in presenting the facts of the case in what appears to the parties as being an unbiased way. There is thus an incentive that the facts are characterised in a way that does not (e.g.) hide certain crucial events or misrepresent them. So this, again, is an argument in support of the hypothesis that the chunks of text found in judgments can indeed be reasonably taken to stand as crude proxies for other kinds of texts (lodged applications/briefs of parties), i.e. to the main candidates for textual analysis of ex ante predictions of outcomes to which we do not have access. Moreover, we fully appreciate the possibility of forms of non-outright bias (and we have revised our text accordingly), but, as already stated at various points, we cannot control for the (eventual) presence of that variable, because we cannot compare the various versions of fact summaries due to lack of data. We have revised our text accordingly to make this clearer.

**4. Third, the absence of disputes before the ECtHR about the facts does not mean that there cannot be different facts emphasized or prioritized by the judges/registry in light of the analysis that they most likely know will follow.**

We agree with the reviewer, and we have revised our text accordingly (see lines 160-162 of the revised text).

**5. Fourth, the authors say 'the "Circumstances" subsection is the closest (even if sometimes crude) proxy we have to a reliable textual representative of the factual background of a case'. Perhaps this is so, but one may wonder about what 'reliable' is worth here without a clearer sense of the model's utility. Surely the facts as summarized in government and/or applicant arguments might be worth a look if the goal is to assist the Court/lawyers in making ex ante predictions about how cases will be decided, even if they are not 'reliable' in the way a peer-reviewed paper might be**

**reliable?**

The reviewer is right, but, as a matter of fact, and as we have already said, we do not have access to these texts and this is the main reason we have used such crude proxies. We submit once again (see our response to the first point) that such access requires other kinds of resources than those we currently have at our disposal. We also want to stress that, in any event, analysis of such further texts, whose obtention requires other kinds of resources, would be a logical step to take once we have *some* evidence, as our article suggests we do, to the effect that an NLP approach can indeed, in principle, be used to predict outcomes with a certain degree of reliability. In any event, we struck out the word 'reliable', which we do not think adds something substantial to our argument, given the caveats already in place.

**6. At several points (line 325, line 340), the ECtHR seems to be referred to as an appellate court. It is not an appellate court. This means, (1) the range of orders and remedies available to the ECtHR are not those of an appellate court, with consequences for its analysis; (2) the ECtHR will frequently be applying different *legal* tests to those applied by the domestic appellate courts (eg, 'was there a violation of Art5 or Art6?' vs 'was the conviction unsafe?'), and (3) therefore different *facts* or different emphasis on those facts may be of interest to the ECtHR than those that were of interest to the domestic court.**

We agree with the reviewer on the 'appellate court' point and we have revised our text accordingly. On the question of facts, see our response to the first point. It is of course possible that different facts may be of interest to the ECtHR (this is always the case when, e.g. the Court uses its fact-finding powers to investigate whether a Convention right was violated at the domestic level, even if such use is rare) but this just restates the point that the relevant sections we used are only crude and imperfect proxies for the facts of the case.

**General comment:** We agree that there are various constraints that curtail the utility of our work. We have quite clearly stated that we do not have access to lodged applications and briefs submitted by parties, nor will we have in the foreseeable future. Under these conditions, we proceeded to a crude emulation of the process, by using instead chunks of text contained in the judgments themselves, that we believe can reasonably stand as crude proxies for the real thing (summaries of facts and legal arguments in applications and briefs) to provide an initial test for an NLP approach. It is precisely, we think, the *success* of this initial test that corroborates the idea that further testing is needed, with recourse to other kinds of data. Again, we stress that access to such data would open avenues for further research and analysis. However that requires the active collaboration with the Court, and therefore a level of trust that exceeds by far what we can work with at the present moment.

**Reviewer 2**

**7. The authors commented during the review process that there are some barriers for accessing the data, from the EctHR portal. Besides, the authors mentioned that accessing cases from domestic courts is not straightforward either. I would like to see a comment regarding data access issues, perhaps a sentence or two in the conclusion section. Are data access issues stopping or slowing down emerging research as the one presented in this paper? Should cases in the EctHR and domestic cases be easily and freely available? If that it the case, is the EctHR making progress in open their repositories for public good?**

Data access issues slow down emerging research indeed. To the best of our knowledge the HUDOC database has not been designed to support large scale access. It is not possible to download the entire dump as in other large repositories such as Wikipedia. Of course making the data available easily accessible will further enable other studies considering the vast amounts of text and the rich associated metadata contained in HUDOC. We have added a couple of sentences in the conclusions section discussing the issue.

---

## Round 0.4 · accepted · Accept

I am happy to see the improved manuscript and appreciate the efforts from both the authors and reviewer throughout several rounds of revision.

Congratulations!

Reviewer 1 ·

Basic reporting

Fine.

Experimental design

Fine.

Validity of the findings

Fine.

Additional comments

Fine.

---

## Author Rebuttal · Round 0.4

We would like to thank the editor and the reviewers for their constructive comments. See below our point-by-point response.

**Reviewer 1**

**The caveats in the rebuttal and the changes made to the paper are interesting. The relevant limitations that are acknowledged -- those that make the paper reliant on crude proxies -- mean that the paper's arguments would struggle to be satisfactory in a legal paper or in legal argument. I understand that the methods adopted in this area may be different.**

**Four specific revisions:**

**(1) The the rule on exhaustion of domestic remedies seems to be poorly articulated, or perhaps misunderstood, at line 152. In particular, its application at the domestic level seems back-to-front, at least as it is explained here. It should be revised.**

The reviewer is right that maybe the formulation we used was not as clear as it could be. The formulation is this: 'While domestic courts do not necessarily hear complaints on the same legal issues as the ECtHR does, by virtue of the rule of exhaustion of domestic remedies they typically have powers to issue judgments on ECHR-related issues.' We meant to say that the rule of exhaustion of domestic remedies, which is an aspect of the principle of subsidiarity, guarantees that domestic courts will be the first to hear complaints on ECHR-based issues (among other things). Of course, whether these issues can be formulated before domestic courts in ways that invoke specifically the Convention will depend on whether the ECHR is indeed incorporated and directly applicable before domestic courts. This is currently the case for all States Parties to the Convention. As the Committee of Ministers puts it (Appendix to Recommendation Rec(2004)6 of the Committee of Ministers to Member States on the improvement of domestic remedies (12 May 2004), at 3–4): '[the ECHR] has become an integral part of the domestic legal orders of all states parties'. In any event, we have further clarified the point in the text (see lines 147-149).

**(2) If there is literature to support the assumption made in lines 108-111, that literature should be noted. Or is it the literature in lines 134-173?**

We know of no academic literature to support this claim with regard to pending applications and/or briefs, the situation being different with respect to domestic judgments (where we provided references in lines 134-173). The reason is simply that applications and briefs are not publicly available. However, one of the coauthors has drafted applications to the ECtHR and has handled cases before the Court: his experience was to the effect that the summaries of the facts by the Court bear important similarities to those prepared by individuals and States. The sample of cases, nonetheless, was not significant. Be that as it may, we reiterate that we formulated our premise in a hypothetical way to underscore the fact that more research is

needed, based on the text of applications and briefs, and that the encouraging results we got for the present paper provide us with a reason to proceed to this further research, which we are very keen to undertake in the future.

**(3) The authors have explained eloquently the limitations on their methodology due to unavailable data. It is therefore jarring to see an expression of belief at line 116-117 ("we believe there is") that, surely, is unsupported due to those very limitations? Is belief the basis for scientific argument here?**

We agree with the reviewer and have revised the text accordingly.

**(4) Line 416 says "Large repositories like HUDOC should be easily and freely accessible." This could be more precise: is HUDOC not easily and freely accessible? Isn't the authors' real concern with the fact that HUDOC is a "case law database" and not a "database of case law and other things"?**

We agree with the reviewer and have revised our text accordingly.